# CA-DEL: An Open Multi-Target, Multi-Modal Benchmark for Learning from DNA-Encoded Library Screens

## Abstract

The success of machine learning in drug discovery hinges on learning the relationship between a chemical structure and its biological activity. While DNA-Encoded Library (DEL) technology can generate the massive datasets required for this task, its primary signal—sequencing read counts—is an indirect and often noisy proxy for true molecular binding affinity. To address the scarcity of public benchmarks for developing robust models that can overcome this data challenge, we introduce CA-DEL. CA-DEL is a multi-dimensional public benchmark featuring screens against three homologous carbonic anhydrase isoforms. Notably, it is the first public DEL dataset to integrate both 2D chemical structures and pre-computed 3D protein-ligand conformations. Crucially, CA-DEL includes a validation set with experimentally determined binding affinities ($K_i$ values). This unique feature enables the direct evaluation of a model's ability to predict true biological activity, rather than simply modeling the noisy enrichment signal.

## 1 Introduction

DNA-Encoded Library (DEL) technology has become a cornerstone of modern drug discovery, enabling unprecedented screening throughput of libraries containing billions of unique molecules covalently linked to DNA barcodes (Needels et al., 1993; Brenner & Lerner, 1992). However, the primary experimental output—high-throughput sequencing read counts—is not a direct measure of binding affinity but rather a noisy proxy confounded by non-specific binding, synthesis impurities, and PCR amplification preferences Dumelin et al. (2006); Wichert et al. (2024). Effective DEL analysis therefore requires computational models capable of denoising, debiasing, and ranking molecules from these weak, indirect signals Ma et al. (2021); Lim et al. (2022); Gu et al. (2024); Cao et al. (2024); Iqbal et al. (2025).

Current approaches face three fundamental limitations. First, molecular recognition is inherently three-dimensional, yet most models rely on 2D representations that discard critical geometric and stereochemical information Shmilovich et al. (2023). This abstraction imposes a ceiling on predictive accuracy when subtle structural differences dictate binding. Second, existing benchmarks suffer from significant constraints: KinDEL covers only two kinase targets with measurement uncertainties Chen et al. (2024), while BELKA, despite containing 133M molecules, lacks fine-grained affinity information and systematic 3D structural information essential for geometric modeling Quigley et al. (2024). Third, the clinical imperative extends beyond finding binders to identifying therapeutically useful molecules with high target selectivity—a challenge particularly demanding for highly homologous protein families where conserved mechanisms make selective inhibitor design difficult.

To address these limitations, we introduce CA-DEL, a multi-dimensional public benchmark dataset designed specifically for DEL data analysis that advances beyond existing resources in scope, modality, and biological relevance. Our contributions include:

- **Multi-target selectivity benchmark.** We evaluate selectivity prediction against highly homologous carbonic anhydrase isoforms (CAII, CAIX, CAXII), representing a clinically relevant challenge where inhibitor selectivity must be achieved between isoforms with conserved catalytic mechanisms.

- **Multi-modal molecular representations.** We integrate traditional 2D molecular topology with systematically generated 3D protein-ligand conformations for over 200K compounds, enabling a large-scale evaluation of geometric deep learning approaches on CA DEL data.

- **Ground-truth validated OOD challenge.** We design an Out-of-Distribution task that tests generalization from noisy DEL screening data (enrichment factors) to precise ChEMBL binding affinities ($K_i/K_d$ values), spanning distinct chemical spaces that mirror real-world hit-to-lead optimization.

- **Practical evaluation metrics.** We introduce Top-N hit rate analysis that directly assesses model utility in resource-constrained discovery campaigns, moving beyond traditional correlation metrics to measure practical discovery value.

Unlike existing benchmarks that focus primarily on binary classification or single-target activity prediction, CA-DEL provides a comprehensive platform for developing and evaluating 3D-aware, selectivity-focused machine learning models on DEL data, potentially catalyzing advances in geometric deep learning approaches for structure-based drug design.

This benchmark design addresses the clinical reality where models trained on initial screening data must generalize to lead-optimized compounds occupying different chemical space, while forcing models to learn isoform-specific binding features rather than generalized motifs. The Top-N evaluation paradigm measures what matters most to discovery scientists: the percentage of true hits among top-ranked compounds.

## 2 RELATED WORK

### 2.1 DEL ANALYSIS

The successful application of DEL technology has generated vast quantities of high-dimensional, high-noise screening data, thereby driving the continuous evolution of computational methods. Early approaches focused on hit identification and initial structure-activity relationship (SAR) analysis, relying on methods including physics-based molecular docking simulations (Jiang et al., 2015; Wang et al., 2015), quantitative structure-activity relationship (QSAR) models (Martin et al., 2017), and metrics specific to DEL enrichment such as data aggregation (Satz, 2016), Enrichment Factor calculation, and standardized z-scores (Faver et al., 2019). While intuitive, these methods possess limited capabilities in handling complex, non-linear structure-activity relationships. To overcome these limitations, machine learning models such as gradient boosting machines, random forests, and support vector machines (McCloskey et al., 2020; Stokes et al., 2020; Li et al., 2018; Ballester & Mitchell, 2010) elevated DEL data into a powerful platform for driving predictive modeling. More recently, the rise of deep learning, particularly the application of Graph Neural Networks (GNNs) (Stokes et al., 2020; Ma et al., 2021), has further enhanced the modeling of complex molecular structures.

Nevertheless, the aforementioned methods predominantly rely on 2D topological information. This represents a fundamental limitation, as the physical essence of molecular recognition occurs in three-dimensional space. Consequently, the effective integration of 3D structural information into predictive models has emerged as a key research frontier. Recent innovations include DEL-Dock, a multi-modal neural network that combines 3D conformational information with 2D topology (Shmilovich et al., 2023); sparse learning methods that address noise originating from truncated products and sequencing errors (Kómár & Kalinic, 2020); and DEL-ranking (Cao et al., 2024), which leverages multiple 3D conformations through a designed ranking-based loss function to more effectively correct read count distributions while concurrently addressing issues of distributional noise and shift.

### 2.2 EXISTING BENCHMARKS FOR DEL ANALYSIS

However, the development of advanced algorithms has long been constrained by a core bottleneck: the scarcity of public benchmark datasets. This situation severely restricts the development and fair comparison of novel computational methods. Early public datasets were modest in scale, such as those for casein kinase and an early dataset for CA (Ballester & Mitchell, 2010; Iqbal et al., 2025). They were instrumental in validating specific analytical methods like probabilistic loss functions, demonstrating the significant value of specialized datasets in catalyzing targeted methodological advancements. Subsequently, the advent of large-scale benchmarks propelled the field to new heights.

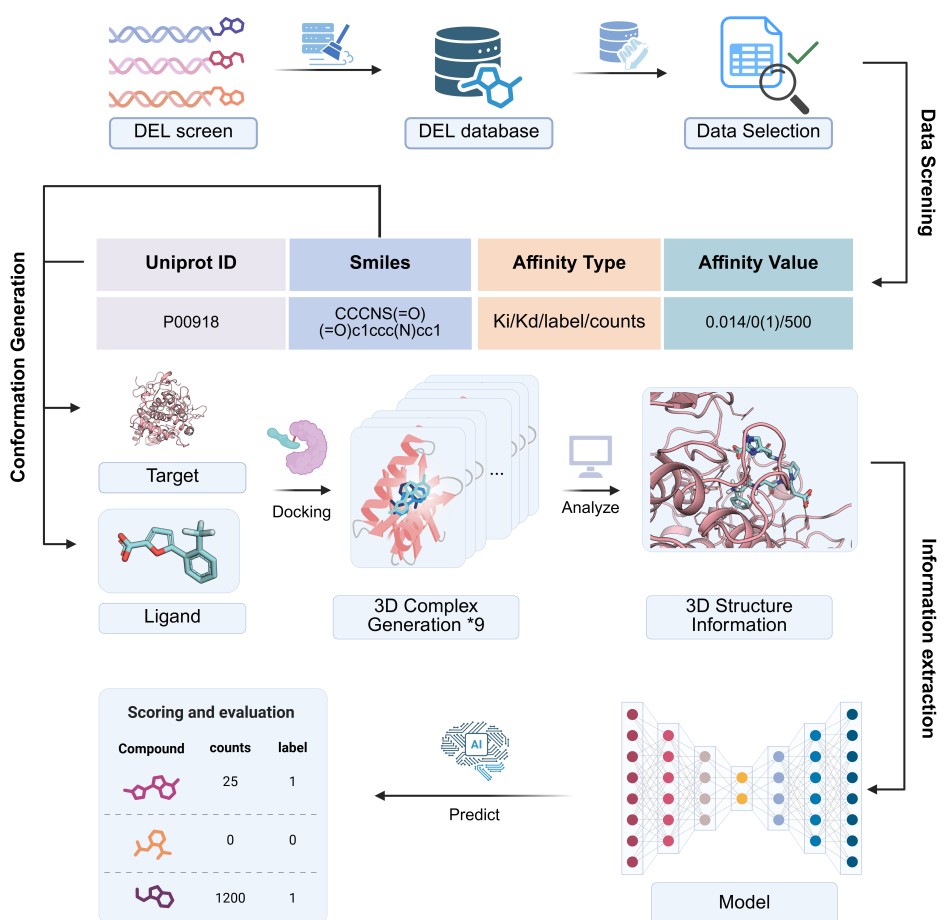

Figure 1: **Schematic overview of the proposed structure-based deep learning framework for DEL analysis.** The workflow is organized into three distinct phases: (1) **Data Screening**, where raw DEL screening data is processed to align target Uniprot IDs with ligand SMILES and affinity metrics; (2) **Conformation Generation**, which utilizes molecular docking to generate 3D protein-ligand complexes. An intermediate *analysis* step is employed to evaluate the docking results and verify the correctness of the binding conformations before finalizing the 3D structure information; and (3) **Information Extraction**, where the validated structural features are fed into a neural network model to predict bioactivity scores and evaluate compound labels.

KinDEL(Chen et al., 2024), one of the first large-scale public datasets, provided data for over 80 million compounds against two kinase targets (MAPK14 and DDR1), including both on-DNA enrichment data and off-DNA biophysical validation data, offering a valuable resource for studying the concordance between DEL screening and traditional assays. More recently, the BELKA dataset, released as part of a Kaggle competition, comprised approximately 133 million molecules against three targets (BRD4, sEH, and HSA). By formulating the problem as a binary classification task (binding vs. non-binding), it significantly lowered the barrier to entry for the broader machine learning community, catalyzing the application of a diverse range of classification algorithms to the DEL hit-finding problem. Current benchmarks for DEL analysis generally lack fine-grained 3D structural information and unified benchmark tasks. This limitation makes it difficult to train and compare models that aim to leverage spatial information for improved predictive accuracy.

## 3 DATASET CONSTRUCTION AND DESCRIPTION

This section details the construction methodology of the CA-DEL dataset (Figure 1), designed to establish a benchmark for evaluating machine learning models on selective protein-ligand binding prediction. We describe target protein selection, small molecule data curation from heterogeneous sources, multi-modal molecular representation generation protocols, and statistical properties of the final dataset.

Table 1: **Overview of the CA-DEL dataset composition.** The benchmark includes three Carbonic Anhydrase isoforms (CAII, CAIX, CAXII). The training sets are derived from large-scale DNA-Encoded Library (DEL) selections containing enrichment factors and binary labels, while the test sets consist of high-quality bioactivity data ($K_i$) curated from ChEMBL for robust evaluation.

| Target | Uniprot ID | PDB ID | Split | Source | Number of Compounds | Data Type | Purpose |
|--------|-----------|--------|-------|--------|---------------------|-----------|---------|
| CAII | P00918 | 3p3h, 5doh | Train | CAS-DEL | 127 500 | Enrichment Data/Label | Training |
| CAIX | Q16790 | 2hkf, 5fl4 | Train | DOS-DEL-1 | 108 528 | Enrichment Data | Training |
| CAXII | O43570 | 4kp5, 4ht2 | Train | CAS-DEL | 127 500 | Enrichment Data/Label | Training |
| CAII | P00918 | 3p3h, 5doh | Test | ChEMBL | 6396 | $K_i$ | Evaluation |
| CAIX | Q16790 | 2hkf, 5fl4 | Test | ChEMBL | 3323 | $K_i$ | Evaluation |
| CAXII | O43570 | 4kp5, 4ht2 | Test | ChEMBL | 2689 | $K_i$ | Evaluation |

### 3.1 TARGET PROTEIN AND SMALL MOLECULE SELECTION

CA-DEL evaluates model generalization across two critical dimensions: *biological selectivity* between homologous targets and *domain generalization* across data distributions. We selected three human carbonic anhydrase isoforms: CAII (ubiquitous anti-target), CAIX and CAXII (cancer-specific targets). Despite high active-site homology, these proteins differ significantly in physiological roles, creating a challenging multi-task learning objective where models must learn fine-grained structural features governing isoform selectivity.

The dataset employs heterogeneous data sources to probe model robustness. Training data originates from two DELs with distinct chemical spaces: CAS-DEL library (127,500 compounds) (Hou et al., 2023) for CAII/CAXII and DOS-DEL-1 library (108,528 compounds) (Gerry et al., 2019) for CAIX (Table 1). Validation and test sets source entirely from ChEMBL (Gaulton et al., 2012), comprising drug-like molecules (CAII: 6,396; CAIX: 3,323; CAXII: 2,689) with precise $K_i$ measurements. This deliberate distributional shift—spanning both chemical space and activity label modality—constitutes a realistic OOD generalization challenge that emulates real-world transfer from high-throughput screening to lead optimization. Data field descriptions are provided in Tables A1, A2, and A3.

### 3.2 GENERATION OF MULTI-MODAL MOLECULAR REPRESENTATIONS

Beyond high-throughput sequencing read counts from DEL wet-lab experiments, we established a systematic pipeline for generating multi-modal representations rich in 3D structural information.

For each target protein, two high-resolution crystal structures were selected from the Protein Data Bank (PDB) (CAII: `3p3h`, `5doh`; CAIX: `2hkf`, `5fl4`; CAXII: `4kp5`, `4ht2`) and subjected to standard preparation protocols using PDB2PQR and PROPKA to account for protein conformational flexibility. For each small molecule, an initial 3D conformation was generated from its SMILES string using RDKit, followed by energy minimization with the MMFF94 force field.

Since docking scoring functions often fail to identify the true binding mode as the top-ranked result, we used SMINA (Koes et al., 2013) to generate ensembles of up to nine plausible binding poses for each ligand-protein pair. This approach increases the probability of capturing the correct binding mode, which typically resides among top-scoring poses. Docking was constrained to a 22.5 Å

cubic search space defined by known co-crystallized ligand positions. By providing pose ensembles, our pipeline becomes more resilient to upstream scoring errors and furnishes more physically complete inputs (Shamsian et al., 2023; Wang et al., 2003; Ferrara et al., 2004). This strategy replaces single, potentially incorrect binding mode estimates with robust discrete approximations of the conformational posterior distribution, enabling models to learn more nuanced and generalizable protein-ligand interaction representations.

## 3.3 DATASET STATISTICS AND ANALYSIS

The CA-DEL dataset was constructed as a rigorous benchmark to address two fundamental challenges in computational drug discovery: learning from noisy (Kuai et al., 2018; Montoya et al., 2025; Li et al., 2018; Lim et al., 2022; Kómár & Kalinic, 2020), high-throughput screening data and generalizing to OOD, drug-like chemical matter (Liu et al., 2021; Tossou et al., 2024; Shi et al., 2025)) The following sections detail the dataset's strategic design and insights from preliminary analysis.

**Significant distributional shift between training and test sets.**

We deliberately introduced a significant distributional shift between the training and test sets to simulate the real-world progression from screening hits to optimized lead compounds. This multifaceted domain gap covers disparities in chemical structure and physicochemical properties. A t-SNE projection (Figure A2) starkly visualizes this structural chasm, showing that the training data (combinatorial libraries) and evaluation sets (ChEMBL) occupy distinct, non-overlapping regions of chemical space. Consequently, the model must learn to extrapolate generalizable biophysical principles rather than interpolate library-specific structural motifs.

This structural divergence is mirrored by a systematic shift in physicochemical properties (Figures 2 and A4). Training molecules resemble initial hits, with lower Quantitative Estimate of Drug-likeness (QED) and higher molecular weights. In contrast, the test sets are more drug-like, with higher QED values and optimized weights. This complex, multi-dimensional gap across various descriptors presents a rigorous test, challenging the model to avoid learning spurious, library-specific correlations.

**Physicochemical Properties Comparison**

Figure 2: Comparison of key physicochemical properties (QED, LogP, and Molecular Weight) across distinct ligand datasets, which includes source datasets (DOS-DEL-1 and CAS-DEL), benchmark dataset KinDEL (Chen et al., 2024), and our curated test sets.The light blue areas mark the 10th and 90th percentiles computed for all the FDA approved oral new chemical entities, as reported by (Shultz, 2018). QED: quantitative estimate of druglikeness (Bickerton et al., 2012).

The model is trained on noisy, long-tailed enrichment factors derived from competitive DEL selection experiments(Figure A3), which are inherently a relative and semi-quantitative measure of binding preference. In stark contrast, the evaluation is performed against precise, absolute biophys-

ical measurements such as binding affinities ($K_i/K_d$) or computationally derived docking scores, whose distributions are shown in Figure 3.

This necessitates that the model not only navigates the covariate shift in molecular features but also performs a complex translation from a noisy, relative experimental signal to a quantitative, absolute biophysical value. Success under these conditions would strongly imply that the model has learned a robust and transferable latent representation that captures the fundamental physics of protein-ligand interactions, effectively distilling the true binding signal from the experimental noise and artifacts inherent in the training data.

### 3.3.1 THE BIOLOGICAL SELECTIVITY CHALLENGE

**Predicting selectivity across highly homologous CA isoforms.** The benchmark targets CA isoforms II, IX, and XII, where CAII represents a ubiquitous anti-target and CAIX/CAXII are validated cancer targets. These proteins exhibit high structural similarity in their active sites, making selective inhibitor design challenging (Mboge et al., 2018; Genis et al., 2009; Alterio et al., 2009). The primary challenge stems from the conserved catalytic zinc ion and subtle structural variations, often single amino acid substitutions, that dictate isoform-specific binding. This framework establishes the task as multi-target representation learning, where effective models must identify minute structural and chemical distinctions determining selectivity. The benchmark directly evaluates a model's capacity to guide lead optimization, where achieving selectivity is essential for therapeutic success.

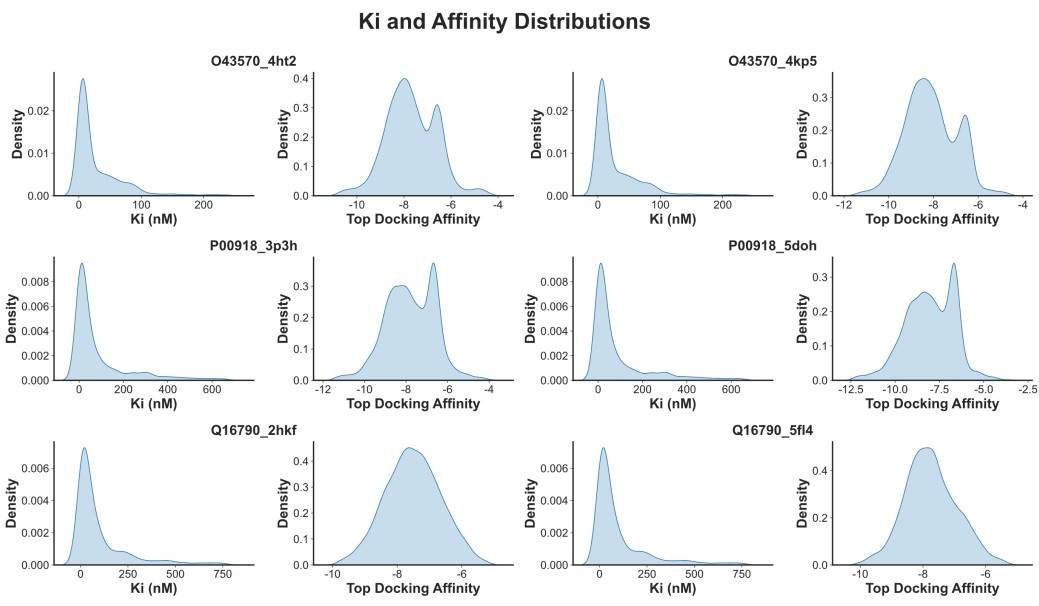

Figure 3: Binding affinity distribution and docking scores for ChEMBL test compounds.

### 3.3.2 GENERATION AND VALIDATION OF PHYSICALLY REALISTIC BINDING POSES

**A cornerstone of our methodology is the generation of physically realistic binding poses whose structural accuracy is rigorously validated against experimental data.** As demonstrated by our re-docking analysis, the generated poses successfully recapitulate the critical binding interactions observed in experimental co-crystal structures (Figure A5). By representing each ligand-protein complex with an ensemble of these validated, low-energy conformations, we decouple model performance from the known unreliability of docking scoring functions (Mukherjee et al., 2010; Scardino et al., 2021; Ferrara et al., 2004). This strategy forces the model to learn "consensus" interaction features that are robust to minor conformational shifts, encouraging the development of representations that better reflect physical reality and exhibit superior generalization.

## 4 RESULTS AND ANALYSIS

To establish a comprehensive performance baseline on the CA-DEL benchmark, we evaluated a range of models, from simple physicochemical descriptors to advanced, multi-modal deep learning architectures. The detailed results are presented in Table 2 and Table 3. Our analysis is structured across multiple dimensions—examining the performance from the perspectives of model methodology, dataset characteristics, and the evaluation metrics themselves—to provide a multi-faceted understanding of the benchmark's challenges and the key drivers of model success.

The primary evaluation task assesses a model's ability to rank compounds by binding potential through correlation with experimental read counts. Performance is measured using Spearman's rank correlation coefficient ($\rho$), defined as:

$$\rho = 1 - \frac{6 \sum_{i=1}^{n} d_i^2}{n(n^2 - 1)} \tag{1}$$

where $d_i$ represents the difference between the ranks of corresponding observations, and $n$ is the number of observations. Based on rank information, we provided Top-N hit rate comparisons to simulate practical utility for scientific discovery and conducted zero-shot generalization experiments for more stringent conditions.

### 4.1 ANALYSIS BY DATASET CHARACTERISTICS AND EVALUATION METRICS

The CA-DEL dataset was intentionally designed to probe fundamental challenges in drug discovery, and the results reflect how different models cope with these hurdles. A primary driver of the observed performance differences is the pronounced Out-of-Distribution (OOD) distributional shift between the DEL-sourced training set and the ChEMBL-sourced test set. Models that merely memorize statistical artifacts of the training library are destined to fail on this task. The poor performance of simple physicochemical baselines and traditional machine learning models on the SubSp metric, which measures correlation with true binding affinity, demonstrates their failure to generalize across this domain shift. In contrast, the success of the 3D deep learning models, such as DEL-Dock and DEL-Ranking, suggests they learn more fundamental and transferable representations of biophysical interactions rather than library-specific patterns. The performance hierarchy among different algorithm types aligns with conventional understanding in computational drug discovery, serving as a crucial validation of the proposed CA-DEL dataset's ability to meaningfully differentiate algorithmic capabilities and capture the inherent complexity of real-world drug discovery scenarios.

Our dual-metric evaluation framework serves as a powerful diagnostic tool for this purpose. Performance is assessed using two Spearman's rank correlation coefficients: Sp, which measures the correlation with the noisy DEL read counts, and SubSp, which measures the correlation with the ground-truth $K_i$ values on the ChEMBL subset. A model that simply overfits to the training data might achieve a high Sp but will fail to generalize, resulting in a low SubSp. Conversely, a model that successfully denoises the training signal will learn the underlying biophysical relationships, resulting in a strong SubSp even if its Sp is not perfect.

**A critical aspect of our evaluation is the interpretation of the SubSp metric.** Since the model is trained to output a higher score for a more promising compound (e.g., from higher enrichment values), while the ground-truth labels are binding affinities like $K_i$ where a *lower* value indicates stronger binding, a strong *negative* correlation (i.e., $\rho \to -1$) signifies superior model performance. This indicates that the model correctly ranks compounds with stronger binding affinity higher.

### 4.2 ANALYSIS BY MODEL COMPLEXITY AND MODALITY

The results reveal a clear performance hierarchy that directly correlates with model sophistication and the nature of the input data in Table 2 and Table 3. This stratification underscores the complexity of the task and highlights the limitations of traditional approaches.

**Physicochemical and Heuristic Baselines.** Simple models utilizing single physicochemical properties, such as Molecular Weight or benzene ring presence, display poor and inconsistent performance across all targets. Their Spearman's rank correlation coefficients for both read counts (Sp) and true

binding affinities (SubSp) approach zero, demonstrating virtually no meaningful predictive capacity for either DEL enrichment signals or genuine biological activity. This demonstrates that simplistic heuristics cannot address the intricate structure-activity relationships embedded within the dataset.

**Classical Docking and Traditional Machine Learning.** The performance of classical molecular docking (Vina Docking) is also limited, serving as a crucial reference point. For instance, on the '5doh' target, its SubSp is merely $-0.017 \pm 0.003$, showing almost no correlation with the true affinities. This result validates our dataset's premise that relying solely on classical docking scores is insufficient and motivates the development of more sophisticated, data-driven models. While traditional machine learning models like Random Forest offer some improvement, they lack the consistency and robustness required for this challenging OOD task.

**Advanced Deep Learning: The Superiority of 3D Representations.** The results unequivocally demonstrate the superiority of deep learning-based approaches, particularly those that integrate 3D structural information. Multi-modal models like DEL-Dock and DEL-Ranking consistently achieve the strongest performance, especially on the critical SubSp metric, which measures correlation with true binding affinity.For the CAIX targets, these models achieve SubSp values as strong as $-0.308$ and $-0.323$ for '2hkf' and '5fl4' respectively. This performance significantly surpasses that of 2D-based models and all other baselines, highlighting the critical importance of leveraging 3D protein-ligand interaction information to successfully generalize from noisy DEL data to OOD chemical space of the ChEMBL test set.

Table 2: Performance of Baseline Models (Part 1). Targets include Carbonic Anhydrase II (CAII, Uniprot: P00918, PDB: 5doh, 3p3h) and Carbonic Anhydrase XII (CAXII, Uniprot: O43570, PDB: 4kp5). The primary metric is Spearman's $\rho$ (Sp) against read counts; the secondary is Spearman's $\rho$ on the subset with ground-truth affinity (SubSp).

| Metric | 5doh | | 3p3h | | 4kp5_A | | 4kp5_OA | |
|---|---|---|---|---|---|---|---|---|
| | Sp | SubSp | Sp | SubSp | Sp | SubSp | Sp | SubSp |
| Mol Weight | -0.250 | -0.125 | -0.250 | -0.125 | -0.101 | 0.020 | -0.101 | 0.020 |
| Benzene | 0.022 | 0.072 | 0.022 | 0.072 | -0.054 | 0.035 | -0.054 | 0.035 |
| Vina Docking | $-0.174_{\pm 0.002}$ | $-0.017_{\pm 0.003}$ | $-0.174_{\pm 0.002}$ | $-0.017_{\pm 0.002}$ | $0.025_{\pm 0.001}$ | $0.150_{\pm 0.003}$ | $0.025_{\pm 0.001}$ | $0.150_{\pm 0.003}$ |
| RF-Enrichment | $0.108_{\pm 0.006}$ | $0.100_{\pm 0.015}$ | $-0.017_{\pm 0.026}$ | $-0.042_{\pm 0.025}$ | $-0.029_{\pm 0.038}$ | $-0.005_{\pm 0.048}$ | $-0.101_{\pm 0.009}$ | $-0.087_{\pm 0.010}$ |
| RF-ZIP | $-0.018_{\pm 0.051}$ | $-0.023_{\pm 0.016}$ | $0.027_{\pm 0.139}$ | $-0.005_{\pm 0.071}$ | $0.035_{\pm 0.094}$ | $-0.026_{\pm 0.111}$ | $0.006_{\pm 0.095}$ | $-0.021_{\pm 0.122}$ |
| Dos-DEL | $-0.053_{\pm 0.019}$ | $-0.012_{\pm 0.027}$ | $-0.048_{\pm 0.036}$ | $-0.011_{\pm 0.035}$ | $-0.016_{\pm 0.029}$ | $-0.017_{\pm 0.021}$ | $-0.003_{\pm 0.030}$ | $-0.048_{\pm 0.034}$ |
| DEL-QSVR | $-0.175_{\pm 0.021}$ | $-0.092_{\pm 0.033}$ | $-0.228_{\pm 0.021}$ | $-0.171_{\pm 0.033}$ | $-0.004_{\pm 0.178}$ | $0.018_{\pm 0.139}$ | $0.070_{\pm 0.0134}$ | $-0.076_{\pm 0.116}$ |
| DEL-Dock | $-0.181_{\pm 0.075}$ | $-0.085_{\pm 0.061}$ | $-0.255_{\pm 0.009}$ | $-0.137_{\pm 0.012}$ | $-0.242_{\pm 0.011}$ | $-0.263_{\pm 0.012}$ | $0.015_{\pm 0.029}$ | $-0.105_{\pm 0.034}$ |
| DEL-Ranking | $-0.262_{\pm 0.013}$ | $-0.140_{\pm 0.021}$ | $-0.286_{\pm 0.002}$ | $-0.177_{\pm 0.005}$ | $-0.268_{\pm 0.012}$ | $-0.277_{\pm 0.016}$ | $-0.289_{\pm 0.025}$ | $-0.233_{\pm 0.021}$ |

Table 3: Performance of Baseline Models (Part 2). Targets include Carbonic Anhydrase XII (CAXII, Uniprot: O43570, PDB: 4ht2) and Carbonic Anhydrase IX (CAIX, Uniprot: Q16790, PDB: 2hkf, 5fl4). The primary metric is Spearman's $\rho$ (Sp) against read counts; the secondary is Spearman's $\rho$ on the subset with ground-truth affinity (SubSp).

| Metric | 4ht2_A | | 4ht2_OA | | 2hkf | | 5fl4 | |
|---|---|---|---|---|---|---|---|---|
| | Sp | SubSp | Sp | SubSp | Sp | SubSp | Sp | SubSp |
| Mol Weight | -0.101 | 0.02 | -0.101 | 0.02 | -0.121 | -0.028 | -0.121 | -0.028 |
| Benzene | -0.054 | 0.035 | -0.054 | 0.035 | -0.174 | -0.134 | -0.174 | -0.134 |
| Vina Docking | $-0.037_{\pm 0.011}$ | $0.092_{\pm 0.011}$ | $-0.037_{\pm 0.011}$ | $0.092_{\pm 0.011}$ | $-0.114_{\pm 0.009}$ | $-0.055_{\pm 0.007}$ | $-0.114_{\pm 0.007}$ | $-0.055_{\pm 0.006}$ |
| RF-Enrichment | $0.011_{\pm 0.027}$ | $0.066_{\pm 0.042}$ | $-0.102_{\pm 0.110}$ | $-0.169_{\pm 0.083}$ | $-0.016_{\pm 0.021}$ | $-0.014_{\pm 0.030}$ | $-0.064_{\pm 0.126}$ | $-0.144_{\pm 0.024}$ |
| RF-ZIP | $-0.265_{\pm 0.014}$ | $-0.222_{\pm 0.018}$ | $0.016_{\pm 0.000}$ | $0.019_{\pm 0.000}$ | $-0.053_{\pm 0.017}$ | $-0.066_{\pm 0.034}$ | $0.040_{\pm 0.022}$ | $-0.011_{\pm 0.042}$ |
| Dos-DEL | $-0.048_{\pm 0.036}$ | $-0.011_{\pm 0.035}$ | $-0.016_{\pm 0.029}$ | $-0.017_{\pm 0.021}$ | $-0.003_{\pm 0.030}$ | $-0.048_{\pm 0.034}$ | $-0.115_{\pm 0.065}$ | $-0.036_{\pm 0.010}$ |
| DEL-QSVR | $-0.228_{\pm 0.021}$ | $-0.171_{\pm 0.033}$ | $-0.004_{\pm 0.178}$ | $0.018_{\pm 0.139}$ | $0.070_{\pm 0.134}$ | $-0.076_{\pm 0.116}$ | $-0.086_{\pm 0.060}$ | $-0.036_{\pm 0.074}$ |
| DEL-Dock | $-0.281_{\pm 0.025}$ | $-0.266_{\pm 0.019}$ | $-0.171_{\pm 0.051}$ | $-0.181_{\pm 0.047}$ | $-0.187_{\pm 0.006}$ | $-0.173_{\pm 0.010}$ | $-0.308_{\pm 0.000}$ | $-0.169_{\pm 0.000}$ |
| DEL-Ranking | $-0.289_{\pm 0.012}$ | $-0.245_{\pm 0.009}$ | $-0.177_{\pm 0.034}$ | $-0.193_{\pm 0.027}$ | $-0.190_{\pm 0.005}$ | $-0.155_{\pm 0.009}$ | $-0.323_{\pm 0.015}$ | $-0.175_{\pm 0.000}$ |

## 4.3 TOP CASE SELECTION

To evaluate a model's practical utility for scientific discovery, we propose the Top-N hit rate as a primary performance metric, defined as the percentage of active compounds ("hits") identified within the top N predictions (Figure 4). By directly quantifying a model's ability to enrich its highest-ranked predictions with active molecules, this metric yields more pragmatically relevant insights than global rank-order statistics: 3D deep learning models demonstrate superior utility. Results

show that models such as DEL-Dock and DEL-Ranking consistently achieve higher hit rates for the most potent binders (top 5% affinity) within the crucial top echelons of the screening list, indicating refined ability to distinguish truly exceptional compounds from merely good ones–a critical function in resource-constrained discovery campaigns. This conclusion derives from simulating realistic scenarios where a model's value depends on its success in concentrating high-quality "hits" for experimental follow-up, rather than correctly ranking the entire library.

The experimental results on the CA-DEL dataset show consistent performance improvements from physicochemical properties to traditional machine learning to deep learning methods. This progression reflects the algorithms' ability to capture increasingly rich information, from basic properties to 2D molecular features to 3D structural data, qualitatively validating the CA-DEL dataset's effectiveness from an information-theoretic perspective.

### 4.4 ZERO-SHOT GENERALIZATION

The zero-shot generalization experiments (Table 4 and Table 5) demonstrate that the predictive performance of models is limited when generalizing across different datasets. This is observed when a model trained on a specific target yields rankings that are poorly correlated with binding potential on an unseen target. In some cases, the correlation becomes positive, indicating a failure in generalization. When the model trained on the 5fl4 dataset was evaluated on the 4kp5_OA target, it produced a Spearman correlation (Sp) of $0.065 \pm 0.021$. A positive correlation in this context suggests that the model's learned patterns are not transferable and lead to systematically incorrect rankings on the new target. This outcome highlights a significant benefit of the CA-DEL benchmark: its capacity to test model robustness against substantial distributional shifts. By exposing the limitations of generalization, the benchmark provides a rigorous framework for assessing whether models are learning fundamental biophysical principles or dataset-specific artifacts, thereby supporting the development of more reliable predictive models.

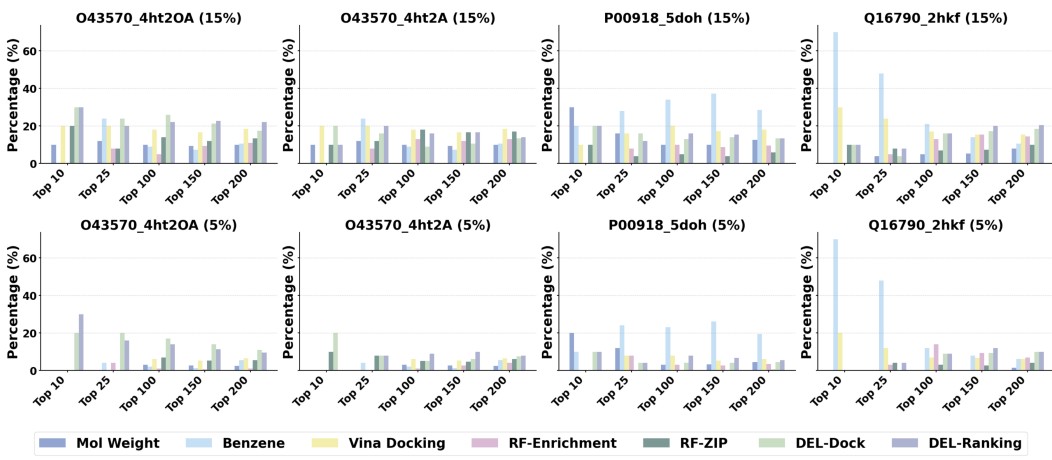

Figure 4: Model performance evaluation using the Top-N hit rate on the CA-DEL dataset. This plot shows the percentage of high-affinity "hits" (defined as the top 5% of binders) successfully identified within the top 200 predictions of each model's ranked list. This metric quantifies the practical utility of a model in resource-constrained screening campaigns.

Table 4: Zero-shot generalization performance of the model trained on the **5fl4** dataset. Scores are evaluated on four distinct test sets.

|  | 3p3h | | 2hkf | | 4kp5_A | | 4kp5_OA | |
|---|---|---|---|---|---|---|---|---|
|  | Sp | SubSp | Sp | SubSp | Sp | SubSp | Sp | SubSp |
| DEL-Dock | $-0.272_{\pm 0.013}$ | $-0.118_{\pm 0.005}$ | $-0.108_{\pm 0.001}$ | $-0.110_{\pm 0.019}$ | $-0.211_{\pm 0.007}$ | $-0.118_{\pm 0.010}$ | $0.065_{\pm 0.021}$ | $-0.125_{\pm 0.034}$ |
| DEL-Ranking | $-0.310_{\pm 0.005}$ | $-0.120_{\pm 0.011}$ | $-0.218_{\pm 0.017}$ | $-0.177_{\pm 0.009}$ | $-0.228_{\pm 0.010}$ | $-0.127_{\pm 0.018}$ | $-0.300_{\pm 0.026}$ | $-0.129_{\pm 0.021}$ |

Table 5: Zero-shot generalization performance of the model trained on the **2hfk** dataset. Scores are evaluated on four distinct test sets.

| | 3p3h | | 5fl4 | | 4kp5_A | | 4kp5_OA | |
|---|---|---|---|---|---|---|---|---|
| | Sp | SubSp | Sp | SubSp | Sp | SubSp | Sp | SubSp |
| DEL-Dock | $-0.185 \pm 0.016$ | $-0.166 \pm 0.008$ | $-0.185 \pm 0.012$ | $-0.162 \pm 0.021$ | $-0.118 \pm 0.009$ | $-0.062 \pm 0.014$ | $0.048 \pm 0.011$ | $0.043 \pm 0.007$ |
| DEL-Ranking | $-0.224 \pm 0.011$ | $-0.209 \pm 0.018$ | $-0.150 \pm 0.015$ | $-0.095 \pm 0.010$ | $-0.174 \pm 0.023$ | $-0.124 \pm 0.006$ | $-0.091 \pm 0.019$ | $-0.044 \pm 0.013$ |

## 5 CONCLUSION

We introduced CA-DEL, a public benchmark dataset to confront core challenges in the machine learning analysis of DEL data. The dataset is multi-target, multi-modal (2D and 3D), and contains experimentally verified binding affinities. Our experiments demonstrate that models leveraging 3D geometric information are more effective at predicting true biological activity from noisy DEL proxy labels. Future work should focus on developing uncertainty-aware models, explainable AI for selectivity, and transfer learning frameworks for new targets.

## 6 ETHICS STATEMENT

This work adheres to the ICLR Code of Ethics. In this study, no human subjects or animal experimentation was involved. All datasets used, including ChEMBL, CAS-DEL, DOS-DEL-1, were sourced in compliance with relevant usage guidelines, ensuring no violation of privacy. We have taken care to avoid any biases or discriminatory outcomes in our research process. No personally identifiable information was used, and no experiments were conducted that could raise privacy or security concerns. We are committed to maintaining transparency and integrity throughout the research process.

## 7 REPRODUCIBILITY STATEMENT

To support the reproducibility of our research, all datasets from this study will be made publicly available. Our implementation builds upon publicly available code from prior work. We have described our key modifications and overall experimental approach in the paper, providing the necessary information to replicate our findings. We believe these measures will enable other researchers to verify our results and build upon our work.

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

## A    APPENDIX

### A.1    LLM USAGE

Large Language Models (LLMs) were used to aid in the writing and polishing of the manuscript. Specifically, we used an LLM to assist in refining the language, improving readability, and ensuring

clarity in various sections of the paper. The model helped with tasks such as sentence rephrasing, grammar checking, and enhancing the overall flow of the text.

It is important to note that the LLM was not involved in the ideation, research methodology, or experimental design. All research concepts, ideas, and analyses were developed and conducted by the authors. The contributions of the LLM were solely focused on improving the linguistic quality of the paper, with no involvement in the scientific content or data analysis.

The authors take full responsibility for the content of the manuscript, including any text generated or polished by the LLM. We have ensured that the LLM-generated text adheres to ethical guidelines and does not contribute to plagiarism or scientific misconduct.

## A.2 APPENDIX FIGURE

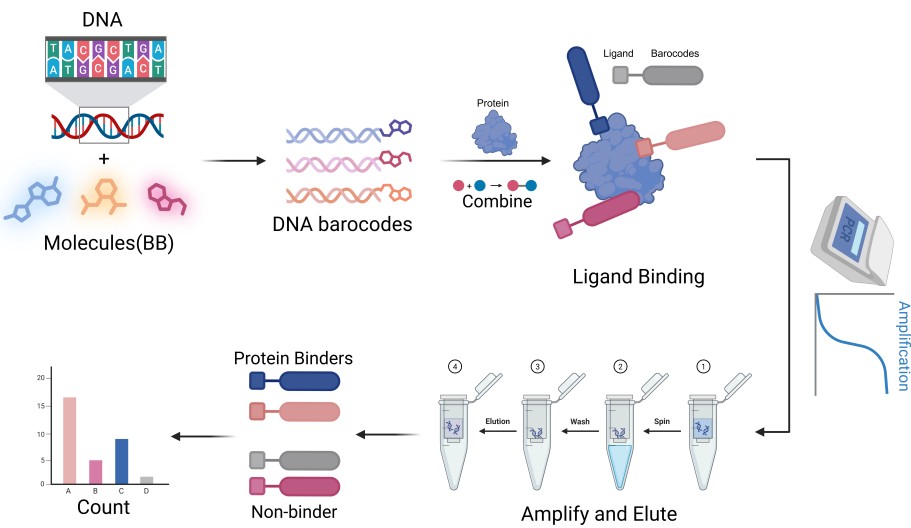

Figure A1: Schematic diagram of the DNA-encoded library (DEL) screening process.

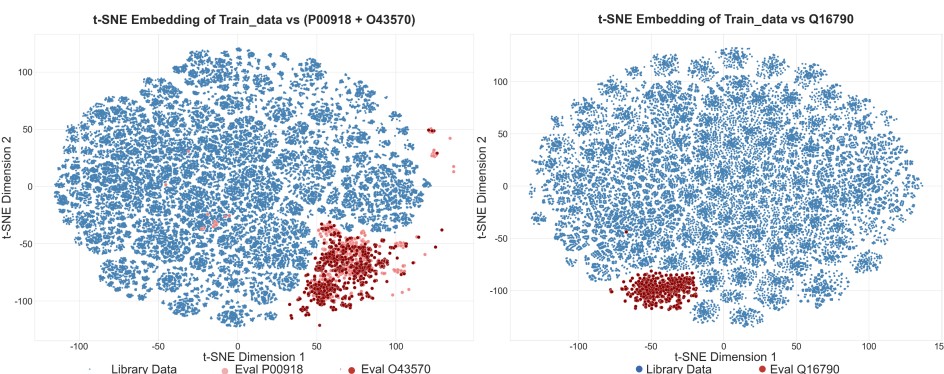

Figure A2: t-SNE visualization of the chemical space. The clear separation between the DEL training set (blue point cloud) and the ChEMBL validation/test set (red star-shaped cluster) highlights the significant distributional shift engineered into the benchmark.

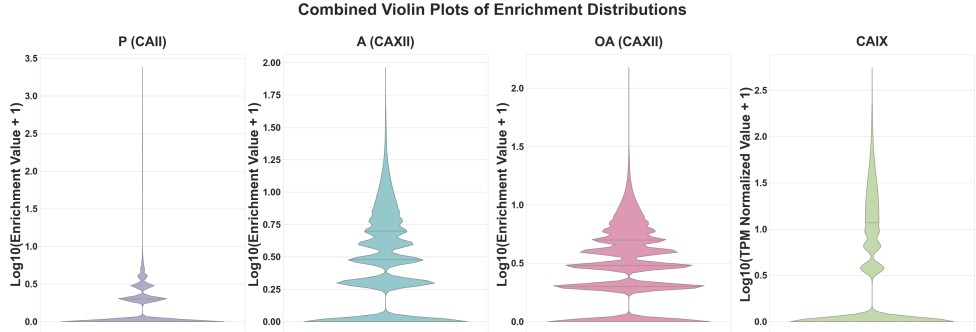

Figure A3: Distribution of enrichment values from the DEL screening data (training set). The violin plot illustrates a typical long-tail distribution, where most library compounds are inactive and a small fraction constitutes potential hits.

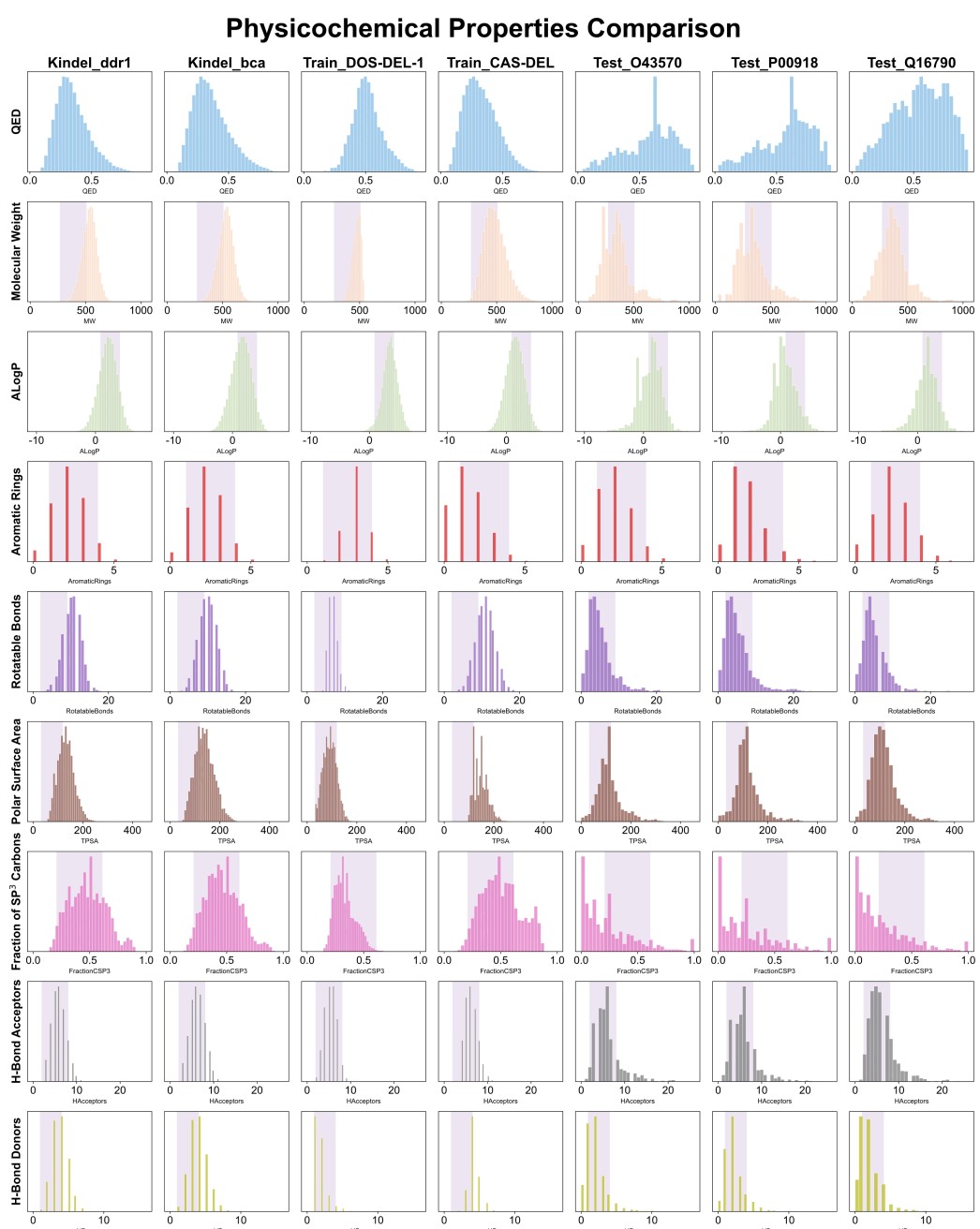

Figure A4: Comparison of key physicochemical properties (QED, LogP, Molecular Weight and others) across distinct ligand datasets, which includes source datasets (DOS-DEL-1 and CAS-DEL), benchmark dataset KinDEL (Chen et al., 2024), and our curated test sets.The light blue areas mark the 10th and 90th percentiles computed for all the FDA approved oral new chemical entities, as reported by (Shultz, 2018). QED: quantitative estimate of druglikeness (Bickerton et al., 2012).

## A.3 DATASET FIELD DESCRIPTIONS

Table A1: Description of data fields in the CAS-DEL dataset files. This table can span multiple pages.

| Field Name | Data Type | Source | Description |
|---|---|---|---|
| smiles | String | CAS-DEL | Simplified Molecular-Input Line-Entry System (SMILES) string representing the final molecule. |
| CodeA, CodeB, CodeC | String / Categorical | CAS-DEL | Unique identifiers for the DNA barcodes or chemical building blocks used in each of the three synthesis rounds. |
| Pre | Integer | CAS-DEL | Pre-selection read counts. |
| P | Integer | CAS-DEL | Raw read counts of DNA tags obtained after the panning step (incubation with the target protein). |
| A | Integer | CAS-DEL | Raw read counts of DNA tags obtained after the amplification step (PCR). |
| OA | Integer | CAS-DEL | Raw read counts of DNA tags obtained after the off-target / counter-selection step. |
| Post | Integer | CAS-DEL | Final enriched DNA tag raw read counts obtained after the complete selection process. |
| label | Integer / Boolean | CAS-DEL | Binary classification label (e.g., 1 for active, 0 for inactive) assigned based on a preset enrichment threshold or validation experiment results. Serves as the target variable for predictive models. |
| row_count | Integer | CAS-DEL | Row index. |
| Exp-B01 | Integer | CAS-DEL | Blank control counts: Raw NGS reads from a control selection performed with blank magnetic beads (no immobilized target protein). |
| Exp-P01, P02, P03 | Integer | CAS-DEL | Post-selection counts (purified protein): Raw NGS reads from three biological replicates of selections against purified, immobilized Carbonic Anhydrase II (CAII). |

Table A1: (continued) Description of data fields in the CAS-DEL dataset files.

| Field Name | Data Type | Source | Description |
| --- | --- | --- | --- |
| Exp-A01, A02, A03 | Integer | CAS-DEL | Post-selection counts (endogenous cellular target): Raw NGS reads from three biological replicates of selections against A549 cells expressing endogenous levels of membrane-bound Carbonic Anhydrase XII (CAXII). |
| Exp-OA01, OA02, OA03 | Integer | CAS-DEL | Post-selection counts (overexpressed cellular target): Raw NGS reads from three biological replicates of selections against hypoxic A549 cells overexpressing Carbonic Anhydrase XII (CAXII). |
| number | Integer | CAS-DEL | Compound serial number. |
| SMILES_BB1, BB2, BB3 | String | CAS-DEL | SMILES strings for the individual Building Blocks (BBs) used to construct the final molecule. |
| BB1D, BB2D, BB3D | String / Categorical | CAS-DEL | Descriptor or ID for the corresponding building blocks. |

Table A2: Description of data fields in the DOS-DEL-1 dataset. This table can span multiple pages.

| Field Name | Data Type | Source | Description |
| --- | --- | --- | --- |
| (Unnamed: 0) | Integer | DOS-DEL-1 | Row index. |
| cpd_id | String / Integer | DOS-DEL-1 | Unique identifier for each compound. |
| scaffold, BB1, BB2 | String / Integer | DOS-DEL-1 | Identifiers for the chemical scaffold and the two variable building blocks (BB1, BB2) used in the DOS-DEL library synthesis. |
| ap1_baseline | Integer | DOS-DEL-1 | Baseline read counts; initial sequencing counts for a CAIX-specific cell line. |
| hrp_beads_r1...r4 | Integer | DOS-DEL-1 | Counter-selection / off-target counts: Raw NGS reads from replicate selections against an unrelated protein (Horseradish Peroxidase, HRP). |

Table A2: (continued) Description of data fields in the DOS-DEL-1 dataset.

| Field Name | Data Type | Source | Description |
|---|---|---|---|
| ca9_beads_r1...r2 | Integer | DOS-DEL-1 | On-target selection counts: Raw NGS reads from replicate selections against the immobilized target protein, Carbonic Anhydrase IX (CAIX). |
| hrp_exp_r1...r2 | Integer | DOS-DEL-1 | Raw NGS reads from replicate experimental output pools in the HRP counter-selection experiment. |
| ca9_exp_r1...r4 | Integer | DOS-DEL-1 | Raw NGS reads from replicate experimental output pools in the CAIX on-target selection experiment. |
| hrp_B, A, Bp, Ap | Float | DOS-DEL-1 | Calculated metrics for HRP counter-selection. Represents: Baseline (B), After-selection (A), Baseline percentage (Bp), and After-selection percentage (Ap). |
| ca9_B, A, Bp, Ap | Float | DOS-DEL-1 | Calculated metrics for CAIX on-target selection, with meanings as described above. |
| ca9_Fn, hrp_Fn | Float | DOS-DEL-1 | Normalized fold-change (Fn): Statistically corrected enrichment scores calculated for the on-target (CAIX) and counter-selection (HRP) screens, respectively. |
| scaffold_smiles, ... | String | DOS-DEL-1 | SMILES strings for the chemical scaffold, building blocks, and the final combined molecule. |
| collection, type | String / Cat. | DOS-DEL-1 | Collection or type metadata for the chemical library. |
| ecfp6 | String / Bit Vector | DOS-DEL-1 | Extended-Connectivity Fingerprint (diameter 6). A hashed numerical representation of molecular structure. |
| cycle0, cycle1, cycle2 | Integer | DOS-DEL-1 | Post-selection round counts: Raw NGS reads after zero, one, or two cycles of selection-amplification. |
| library_id | String / Integer | DOS-DEL-1 | Identifier for the DNA-Encoded Library used. |

Table A2: (continued) Description of data fields in the DOS-DEL-1 dataset.

| Field Name | Data Type | Source | Description |
|---|---|---|---|
| cycle01, cycle02, cycle12 | Float | DOS-DEL-1 | Enrichment ratios between different selection rounds (e.g., cycle1/cycle0), providing a direct measure of a compound's enrichment efficiency. |

Table A3: Description of data fields for the ChEMBL validation and test sets.

| Field Name | Data Type | Source | Description |
|---|---|---|---|
| num | Integer | ChEMBL | Row index. |
| smiles | String | ChEMBL | Simplified Molecular-Input Line-Entry System (SMILES) string. |
| affinity | Float / String | ChEMBL | The specific type of affinity measurement reported (e.g., Ki, Kd, IC50). |
| Ki (nM) | Float | ChEMBL | Inhibition constant, reported in nanomolar (nM) units. |
| Kd (nM) | Float | ChEMBL | Dissociation constant, reported in nanomolar (nM) units. |
| IC50 (nM) | Float | ChEMBL | Half-maximal inhibitory concentration, reported in nanomolar (nM) units. |

## A.4 DETAILED PROTOCOLS AND EXPERIMENTAL SETUP

To facilitate fair and reproducible evaluation on our CA-DEL dataset, we have designed a comprehensive benchmark suite. It comprises four distinct tasks that probe model capabilities across key dimensions, from practical hit identification to advanced QSAR and generalization challenges.

- **Read Count Ranking and Affinity Correlation.** This is the core task of the benchmark, designed to evaluate a model's ability to learn a function from molecular representations that can effectively rank compounds by their potential activity. In the early stages of drug discovery, accurately ranking the most promising compounds is far more critical than predicting their precise activity values. The objective of this task is to train a model that takes molecular information (2D and/or 3D representations) as input and outputs a continuous, real-valued ranking score that is monotonically correlated with the compound's binding potential. Performance is measured by the Spearman's rank correlation coefficient (Spearman's $\rho$) on two levels:

    - *Count-based $\rho$*: The correlation between the model's predicted scores and the experimental sequencing read counts on the entire test set, assessing the model's ability to fit the experimental proxy signal.
    - *Affinity-based $\rho$*: The correlation between the predicted scores and the true binding affinities on the validation subset containing ground-truth labels. This serves as a more stringent measure of the model's ability to identify genuinely active compounds.

- **Hit Compound Classification.** This task simplifies the complex ranking problem into a binary classification problem, simulating the decision-making process of rapidly triaging compounds into "hits" or "non-hits" during a screening campaign. The objective is to train a classifier to distinguish between compounds with high and low read counts. We generate binary labels using a threshold-based method; for instance, compounds in the top 5% of read counts in the test set, or those containing certain key functional groups (e.g., sulfonamides), are defined as the positive class (1), with the rest as the negative class (0). The model takes the 2D and/or 3D representation of a molecule as input and outputs the probability of it belonging to the positive class. Performance is measured by the Area Under the Receiver Operating Characteristic curve (AUC-ROC).

- **Quantitative Affinity Prediction.** This is a more challenging regression task that focuses on leveraging the limited amount of ground-truth activity data to evaluate a model's capability for Quantitative Structure-Activity Relationship (QSAR) modeling. The objective is to train or fine-tune a model on the validation subset containing true $K_i/K_d$ values to directly predict the binding affinity of a compound. The model takes molecular representations as input and outputs a predicted affinity value (typically on a logarithmic scale such as $pK_i$). Performance is evaluated jointly by the Pearson correlation coefficient (Pearson's r) and the Root Mean Square Error (RMSE) to measure the consistency and error magnitude between predicted and true values.

- **Cross-Target Generalization.** This task aims to evaluate a model's ability to learn transferable chemical knowledge by applying what is learned from one or more related targets to a zero-shot prediction scenario on a novel target. This is crucial for assessing a model's generalization capabilities and its potential for application in data-scarce scenarios. The objective is to evaluate the model's performance in ranking activities for a target without any exposure to its data during training. The specific protocol is as follows: a model is trained using all data from the CAII and CAXII targets, and the trained model is then directly applied to make predictions on the test set for the CA IX target without any fine-tuning. The model takes the representations of molecules from the CA IX test set as input and outputs a ranking score for each. Performance is again measured by the Spearman's rank correlation coefficient ($\rho$), calculated between the model's predicted scores and the experimental read counts on the CA IX test set.

To ensure the fairness of all comparisons and the reproducibility of results, we provide standard, fixed data splits for CA-DEL (in `.npz` format). For each sub-dataset, we provide splits for training (80%), validation (10%), and testing (10%). We strongly encourage researchers to adhere to these prescribed splits. All model hyperparameters should be tuned on the validation set, with final performance reported on the independent test set. All code for data processing, task evaluation, and baseline model implementation will be publicly released to support this protocol.

### A.5 MODEL SIZE AND PARAMETERIZATION DETAILS

In this section, we present the relevant hyperparameters. All hyperparameters were tuned to achieve optimal performance on their respective datasets.

Table A4: DEL-Docking Hyperparameters (Model Structure Parameters)

| Dataset | learning_rate | lrd_gamma | n_layers | dropout |
|---------|---------------|-----------|----------|---------|
| 2hkf    | 3.00E-04      | 0.1       | 2        | 0.5     |
| 3p3h    | 8.00E-06      | 0.1       | 2        | 0.5     |
| 4ht2A   | 2.00E-05      | 0.1       | 2        | 0.5     |
| 4ht2OA  | 2.00E-05      | 0.1       | 2        | 0.5     |
| 4kp5A   | 8.00E-05      | 0.1       | 2        | 0.5     |
| 4kp5OA  | 1.00E-04      | 0.1       | 2        | 0.5     |
| 5doh    | 5.00E-06      | 0.1       | 2        | 0.5     |
| 5fl4    | 1.00E-04      | 0.1       | 2        | 0.5     |

Table A5: DEL-Ranking Hyperparameters (Optimization Parameters)

| Dataset | lr | listmle_weight | $\lambda_w$ | Temperature |
|---|---|---|---|---|
| 2hkf | 8.00E-05 | 1.00E-09 | 1.00E+09 | 0.1 |
| 3p3h | 3.00E-05 | 1.00E-09 | 1.00E+09 | 0.5 |
| 4ht2A | 3.00E-05 | 1.00E-09 | 1.00E+09 | 0.8 |
| 4ht2OA | 3.00E-05 | 1.00E-09 | 1.00E+09 | 0.8 |
| 4kp5A | 1.00E-03 | 1.00E-09 | 1.00E+09 | 0.3 |
| 4kp5OA | 1.00E-03 | 1.00E-10 | 1.00E+10 | 0.2 |
| 5doh | 1.00E-04 | 1.00E-10 | 1.00E+10 | 0.8 |
| 5fl4 | 1.00E-04 | 1.00E-08 | 1.00E+08 | 0.9 |

## A.6 BASELINE MODELS AND EXPERIMENTAL SETUP

To provide a solid performance reference for future research, we established a suite of baseline models spanning various levels of complexity and data modalities. The selection of these models is intended to systematically probe the capabilities and limitations of different methodologies in addressing the unique challenges presented by the CA-DEL dataset. Our library of baseline models extends from simple physicochemical prior-based methods to advanced multi-modal deep learning models, and specifically includes:

- **Training DEL for Ranking Targets:** This task focuses on training the model to distinguish active compounds from decoys, effectively solving the hit-finding problem through classification and ranking.

- **Tuning DEL with $K_i/K_d$ for Targets Affinity Prediction:** Here, the model is fine-tuned for a quantitative regression task to predict the precise binding affinity of molecules to their biological targets.

- **Generalizing DEL for New Targets:** This task evaluates the model's ability to transfer its learned knowledge and make accurate predictions for novel protein targets not seen during training.

All experiments were completed on a computing cluster equipped with NVIDIA A100 GPUs. We specifically note that the computational cost of the 3D models is significantly higher than that of the 2D models due to the need to process multiple 3D conformations for each molecule, a fact that highlights the importance of releasing pre-computed features and standardized benchmarks. Finally, to promote full reproducibility and further research by the community, all associated code will be made publicly available through a GitHub repository.

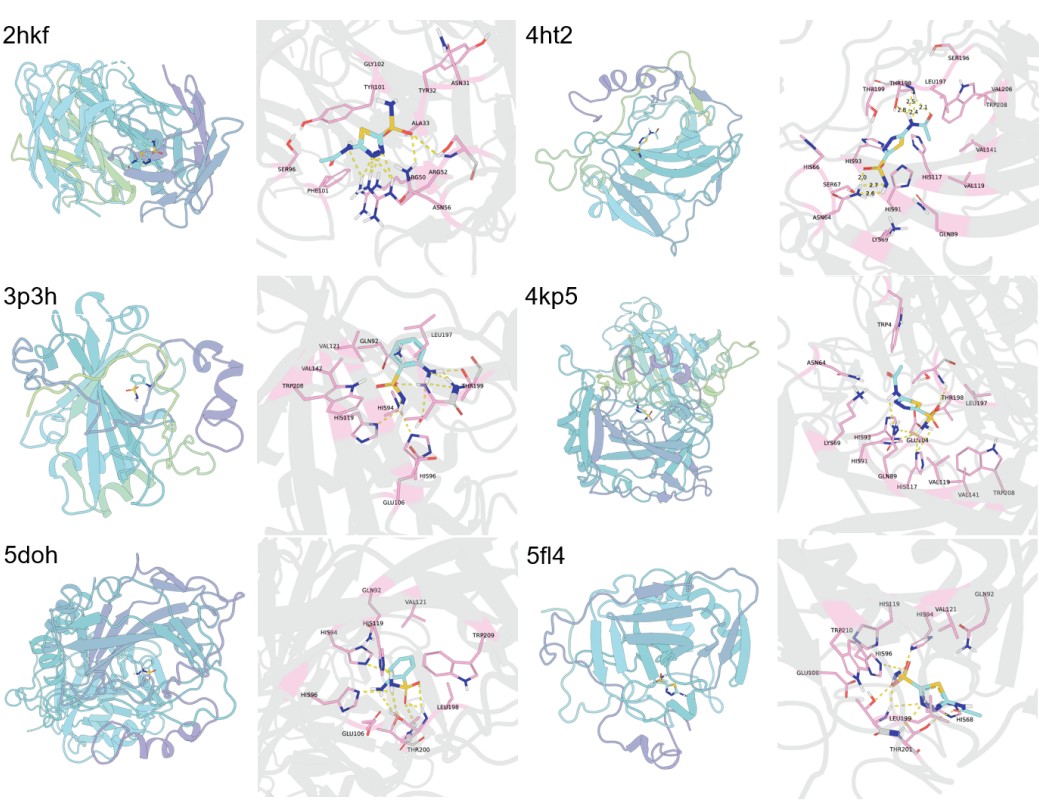

Figure A5: Validation of the docking protocol via re-docking. **(Left)** Overall view of the computationally generated binding pose (ligand shown in cyan sticks) within the protein's active site. **(Right)** A zoomed-in view detailing the critical interactions. The computationally generated pose (cyan) accurately reproduces the experimental binding mode(, such as zinc chelation and hydrogen bonds), forming key interactions with the active site residues (pink sticks).

