# OpenReview forum: "CA-DEL: Construction and Application of an Intelligent DEL Database for Anti-Cancer Drug Discovery"
_ICLR.cc/2026/Conference — ICLR 2026 Conference Desk Rejected Submission_

### Official Review · Reviewer_ra3Z · 2025-10-31

**Soundness:** 2
**Presentation:** 2
**Contribution:** 1
**Rating:** 2
**Confidence:** 3

**Summary:**

The authors introduce a benchmark for protein-target binding for DNA-encoded libray (DEL) data. Specifically, they provide binding data for three highly homologous proteins. They include 2D and 3D molecular data for the compounds provided. They also include a specific data split, where training data is collected from DEL datasets, and the test set is selected from ChEMBL.

**Strengths:**

- A new dataset for drug discovery is provided, specific for  DNA encoded libraries.
- The authors include an anti-target to the benchmark to assess that models have to find very specific targets that bind to the desired target but not similar ones.
- The main figure provides a good overview of the dataset and its intended use.

**Weaknesses:**

- Not enough targets: although the authors point out the KinDEL dataset having only two proteins as a critical limitation, the benchmark proposed in this manuscript provides only three structures. Moreover, all three systems included in this benchmark are isoforms of the carbonic anhydrase. Therefore, this benchmark provides quite narrow possibilites for evaluation.
- The 3D structures provided in this paper come from protein docking, so they are not an experimentally obtained ground truth.
- The authors claim to include a realistic OOD evaluation scenario, because they evaluate on molecules collected from ChEMBL. However, they don't perform any evaluation regarding how similar training and test compounds are, neither a scaffold split is performed to evaluate the models on more structurally different compounds.

**Questions:**

- Why is it necessary to train models in a hit screening setting and evaluate them in a hit to lead optimization setting? Wouldn't it be possible to have a model for the initial hit screening setting and another, more specific model for hit to lead development?
- Why is correlation measured instead of the mean squared error in the test set?
- In Tables 2 and 3, it is hard to relate the protein code to the specific protein tht is being evauated, I recommentd including the protein names and remark which one of them is the anti-target.
- In Table 1, the training sets have "Enrichment Data/Label" as data type. However, it is not explained exactly what this data type consists of. Could you please explain it in more detail?

---

> ### Author Response · Authors · 2025-11-27
> **Rebuttal by Authors**
>
> Thank you very much for your reply! Regarding your concerns, our responses are below:
> ### Weakness 1
>
> **Reviewer Comment:**
> > Not enough targets: although the authors point out the KinDEL dataset having only two proteins as a critical limitation, the benchmark proposed in this manuscript provides only three structures. Moreover, all three systems included in this benchmark are isoforms of the carbonic anhydrase. Therefore, this benchmark provides quite narrow possibilites for evaluation.
>
> **Response:**
> We appreciate this perspective. We believe that both KinDEL and our CA-DEL represent distinct and substantial contributions to the DEL machine learning landscape. While KinDEL provides a critical resource for kinase-focused research, our work pioneers the evaluation of **isoform selectivity** within the Carbonic Anhydrase (CA) family—a more nuanced and structurally complex challenge in drug discovery.
>
> Rather than simply increasing the number of targets, we deliberately focused on high-homology isoforms to test a model's ability to discern subtle structural differences, which is the essence of lead optimization.
>
> Furthermore, we consider CA-DEL as a proof-of-concept framework. The pipeline we established (Enrichment calculation $\rightarrow$ Docking/AF3 $\rightarrow$ Validation) is protein-agnostic. We are actively planning to extend this workflow to broader datasets generated from diverse DEL wet-lab experiments in our future work to further enhance target diversity.
>
> ---
>
> ### Weakness 2
>
> **Reviewer Comment:**
> > The 3D structures provided in this paper come from protein docking, so they are not an experimentally obtained ground truth.
>
> **Response:**
> We fully agree that experimental structures (e.g., Cryo-EM, X-ray) remain the gold standard. However, determining experimental structures for the millions of ligands in a DNA-Encoded Library is currently intractable due to scale.
>
> Our reliance on docking is a strategic choice to bridge this gap. We do not aim to predict the protein's conformational changes (folding), but rather the ligand's binding mode within a known pocket. Given that the CA pocket is well-defined, physics-based docking provides a computationally feasible and scientifically grounded approximation. This allows us to unlock the massive, previously underutilized structural potential of DEL data.
>
> We also note that recent benchmarks (e.g., PoseBusters) suggest that for known pockets, physics-based docking remains highly competitive with methods like AlphaFold 3 for small molecule pose prediction.
>
> ---
>
> ### Weakness 3
>
> **Reviewer Comment:**
> > The authors claim to include a realistic OOD evaluation scenario... However, they don't perform any evaluation regarding how similar training and test compounds are...
>
> **Response:**
> Thank you for raising this important point regarding OOD (Out-of-Distribution) evaluation. We actually addressed the low similarity between the datasets in **Figure A2 (t-SNE distribution)** and **Figure A4**.
>
> * **DEL Training Data:** Constructed from specific synthetic "blocks" (leading to specific chemical spaces).
> * **ChEMBL Test Set:** Represents a diverse history of medicinal chemistry optimization.
>
> As shown in the t-SNE plot (Figure A2), the chemical space distributions of the DEL data and the ChEMBL validation set are distinct, confirming that our evaluation setup inherently tests the model's generalization capability across structurally diverse compounds.
>
> ---
>
> ### Question 1
>
> **Reviewer Comment:**
> > Why is it necessary to train models in a hit screening setting and evaluate them in a hit to lead optimization setting? Wouldn't it be possible to have a model for the initial hit screening setting and another, more specific model for hit to lead development?
>
> **Response:**
> We completely agree that in an industrial pipeline, specialized models are often used. However, the academic objective of this study is to demonstrate a more challenging capability: **Transfer Learning**.
>
> We aim to prove that a model can learn fundamental binding physics from the "Screening Environment" (massively abundant but noisy DEL data) and successfully generalize that knowledge to the "Optimization Environment" (data-scarce but high-precision ChEMBL data). Validating this transferability opens the door to utilizing the vast scale of DEL data to train robust models for targets where optimization data is scarce or unavailable.
>
> ---

---

> > ### Author Response · Authors · 2025-11-27
> > **Rebuttal by Authors**
> >
> > ### Question 2
> >
> > **Reviewer Comment:**
> > > Why is correlation measured instead of the mean squared error in the test set?
> >
> > **Response:**
> > The choice of Spearman correlation ($\rho$) over MSE is dictated by the fundamental nature of this cross-domain study.
> >
> > * **Training Labels:** DEL Enrichment Values (relative, semi-quantitative).
> > * **Test Labels:** ChEMBL $pK_i/pK_d$ (absolute, logarithmic physical constants).
> >
> > Calculating MSE between these disparate units would be scientifically invalid. Our core objective is not to regress the exact absolute $pK_d$ (which the model has never seen), but to verify if the model has learned the **ranking logic**—i.e., can it distinguish a "good" binder from a "bad" one? Spearman correlation is a non-parametric metric that perfectly captures this "consistency of ranking," which aligns with the primary goal of virtual screening.
> >
> > ---
> >
> > ### Question 3
> >
> > **Reviewer Comment:**
> > > In Tables 2 and 3, it is hard to relate the protein code to the specific protein tht is being evauated, I recommentd including the protein names and remark which one of them is the anti-target.
> >
> > **Response:**
> > Thank you for pointing out this clarity issue. We have updated the captions for both **Table 2** and **Table 3** in the revised manuscript to explicitly map the protein codes to their full names and clearly indicate which targets serve as anti-targets (negative controls).
> >
> > ---
> >
> > ### Question 4
> >
> > **Reviewer Comment:**
> > > In Table 1, the training sets have "Enrichment Data/Label" as data type. However, it is not explained exactly what this data type consists of. Could you please explain it in more detail?
> >
> > **Response:**
> > We apologize for the ambiguity. "Enrichment Data/Labels" refers to the specific DEL-derived metrics used for training:
> >
> > * **Enrichment Data:** The calculated enrichment factors derived from the selection counts.
> > * **Binary Labels:** The $0/1$ classification labels used for discriminatory tasks.
> >
> > Comprehensive definitions of these data components are already detailed in **Tables A1–A3** of the Appendix. These tables explicitly describe all raw inputs, such as the "Blank control counts" (raw NGS reads from blank bead selections) used to derive the enrichment metrics.

---

### Official Review · Reviewer_gdgv · 2025-11-01

**Soundness:** 2
**Presentation:** 3
**Contribution:** 2
**Rating:** 4
**Confidence:** 4

**Summary:**

The authors present CA-DEL, an open benchmark for ML on DNA-Encoded Library (DEL) screens that tackles the core challenge of noisy sequencing-based enrichment as a proxy for binding affinity. The dataset targets a clinically relevant selectivity setting across three homologous carbonic anhydrase isoforms (CA-II, CA-IX, CA-XII), where small structural differences matter. CA-DEL couples multi-target tasks with multi-modal inputs—2D SMILES-derived features and large-scale docked 3D protein–ligand poses—and includes out-of-distribution ground-truth affinities from ChEMBL for rigorous validation of biological activity, not just DEL correlation. Baseline comparisons indicate that 3D-aware geometric models outperform 2D/classical methods on both DEL enrichment correlation and true affinity prediction, supporting the premise that explicit structure helps denoise DEL signals and transfer to biochemical endpoints.

**Strengths:**

1) Attempts to address a significant bottleneck in machine learning for drug discovery.
2) CA-DEL is the first open dataset to integrate multi-target DEL data with experimentally validated Ki values and 3D structural information.
3) OOD generalization scheme is novel in DEL evaluation and reflects practical challenges in medicinal chemistry.

**Weaknesses:**

1) The authors introduce and dataset and benchmarking scheme rather than an architectural innovation. The dataset presents the opportunity to develop a 3D structure-aware ligand–protein binding model specializing in generalizing to true binding affinities, but the authors do not explore new architectures along this natural avenue Moreover, the evaluation relies primarily on standard metrics (Spearman correlation and top-K hit rate) without proposing new evaluation criteria. It would be valuable to experiment with a new method for uncertainty modeling or elaborate on the discussion about it.)

2) The authors focus exclusively on a single protein family of proteins (carbonic anhydrase isoforms). This choice is scientifically well-motivated since isoform selectivity is difficult and clinically relevant, but additional diversity across protein families would strengthen the dataset's generalizability and broaden its applicability. Demonstrating that similar construction pipelines could extend to other target classes would strengthen the contribution further.

3) The authors also mentioned noise and bias in the synthetic DEL data and simulated docking. Have the authors conducted any work to determine the extent, influence on the model, and any corrections to this.

4) It would be helpful to provide more information on the models used, specifically model size and parameterization details. The key point in the paper is that training on 3D protein-ligand conformations benefits biological understanding. To further demonstrate the explicit benefit of 3D information, an ablation study on the DEL-Dock and DEL-Ranking models without 3D conformation data would be interesting to see.

**Questions:**

Please address points mentioned in the weakness section.

---

> ### Author Response · Authors · 2025-11-27
> **Rebuttal by Authors**
>
> Thank you very much for your reply! Regarding your concerns, our responses are below:
> ### Weakness 1
>
> **Reviewer Comment:**
> > The authors introduce and dataset and benchmarking scheme rather than an architectural innovation... It would be valuable to experiment with a new method for uncertainty modeling or elaborate on the discussion about it.
>
> **Response:**
> We appreciate your insightful suggestion regarding uncertainty modeling. We fully agree that developing architectures capable of quantifying uncertainty is a high-value direction for the field.
>
> As a submission to the **Datasets and Benchmarks** track, our primary goal was to construct a high-quality, structure-annotated resource that empowers the community to explore these advanced architectures. By providing reliable 3D structural priors, CA-DEL serves as the foundational "fuel" required to train and test the next generation of models, including those focused on uncertainty estimation. We selected Spearman correlation ($Sp$) and Top-K hit rates because they are the established standard metrics in the DEL field. This ensures our benchmark provides a direct and fair comparison with existing literature.
>
>
>
> Taking your suggestion into account, we are currently developing a **Point-Cloud Evolutionary Network**. This model utilizes the 3D structural priors from CA-DEL to initialize geometric embeddings, which are then refined through an evolutionary attention mechanism to dynamically model the uncertainty distribution of ligand-protein interactions.
>
> ---
>
> ### Weakness 2
>
> **Reviewer Comment:**
> > The authors focus exclusively on a single protein family of proteins (carbonic anhydrase isoforms)... additional diversity across protein families would strengthen the dataset's generalizability...
>
> **Response:**
> We share your vision regarding the importance of generalizability and broad applicability. We selected the **Carbonic Anhydrase (CA)** family as a strategic starting point because its isoforms share high structural similarity yet exhibit distinct binding profiles. This makes it an ideal testbed for the critical challenge of selectivity in drug discovery.
>
>
>
> We are pleased to clarify that the CA-DEL construction pipeline is designed as a modular, protein-agnostic framework. The workflow—spanning enrichment calculation, structural generation (via docking or AlphaFold 3), and validation—can be readily applied to other protein families. We agree that extending this to other targets is the natural next step. We plan to validate this expansion using public datasets (such as ChEMBL or those with experimental $K_i$/$K_d$ data) to further demonstrate the robustness of our pipeline in future iterations.
>
> ---

---

> > ### Author Response · Authors · 2025-11-27
> > **Rebuttal by Authors**
> >
> > ### Weakness 3
> >
> > **Reviewer Comment:**
> > > The authors also mentioned noise and bias in the synthetic DEL data and simulated docking. Have the authors conducted any work to determine the extent, influence on the model, and any corrections to this.
> >
> > **Response:**
> > Quantifying the exact degree of noise is indeed challenging due to its multifactorial nature in DEL screens, which stems from stochastic Poisson fluctuations, PCR amplification bias, and variable synthesis yields. Instead of a point-by-point quantification, we addressed this through a combined strategy of **data-level mitigation** and **algorithmic benchmarking**.
> >
> > * **Data Level:** We employed "Blank Control" samples to remove systematic bias via enrichment factor calculation and utilized docking scores as physical constraints to filter false positives that lack structural complementarity.
> > * **Model Influence:** We acknowledge that different algorithms exhibit varying capabilities in handling such intrinsic noise, meaning the "correction" effect is inherently model-dependent.
> >
> > Since the extensive data obtained from DEL experiments makes it infeasible to obtain one-to-one parallel experimental affinity data (e.g., $K_i$ or $K_d$) for validation, our study addresses this by comprehensively benchmarking diverse architectures. This comparative approach allows us to evaluate the relative robustness of different models, effectively identifying which algorithms are most resilient to noise in the absence of complete wet-lab ground truth.
> >
> > ---
> >
> > ### Weakness 4
> >
> > **Reviewer Comment:**
> > > It would be helpful to provide more information on the models used, specifically model size and parameterization details... an ablation study on the DEL-Dock and DEL-Ranking models without 3D conformation data would be interesting to see.
> >
> > **Response:**
> > We present the relevant hyperparameters in **Table A4** and **A5** below:
> >
> > **Table A4: DEL-docking Hyperparameters**
> >
> > | Dataset | learning_rate | lrd_gamma | n_layers | dropout |
> > | :--- | :---: | :---: | :---: | :---: |
> > | 2hkf | 3.00E-04 | 0.1 | 2 | 0.5 |
> > | 3p3h | 8.00E-06 | 0.1 | 2 | 0.5 |
> > | 4ht2A | 2.00E-05 | 0.1 | 2 | 0.5 |
> > | 4ht2OA | 2.00E-05 | 0.1 | 2 | 0.5 |
> > | 4kp5A | 8.00E-05 | 0.1 | 2 | 0.5 |
> > | 4kp5OA | 1.00E-04 | 0.1 | 2 | 0.5 |
> > | 5doh | 5.00E-06 | 0.1 | 2 | 0.5 |
> > | 5fl4 | 1.00E-04 | 0.1 | 2 | 0.5 |
> >
> > **Table A5: DEL-ranking Hyperparameters**
> >
> > | Dataset | lr | listmle_weight | lambda_w | temperature |
> > | :--- | :---: | :---: | :---: | :---: |
> > | 2hkf | 8.00E-05 | 1.00E-09 | 1.00E+09 | 0.1 |
> > | 3p3h | 3.00E-05 | 1.00E-09 | 1.00E+09 | 0.5 |
> > | 4ht2A | 3.00E-05 | 1.00E-09 | 1.00E+09 | 0.8 |
> > | 4ht2OA | 3.00E-05 | 1.00E-09 | 1.00E+09 | 0.8 |
> > | 4kp5A | 1.00E-03 | 1.00E-09 | 1.00E+09 | 0.3 |
> > | 4kp5OA | 1.00E-03 | 1.00E-10 | 1.00E+10 | 0.2 |
> > | 5doh | 1.00E-04 | 1.00E-10 | 1.00E+10 | 0.8 |
> > | 5fl4 | 1.00E-04 | 1.00E-08 | 1.00E+08 | 0.9 |
> >
> > To explicitly demonstrate the value of 3D information, we refer to the comparison in **Table 2**, where structure-aware models (like DEL-Dock) consistently outperform 1D/2D-based baselines (such as MLP-ZIP and Random Forest). This performance gap serves as an empirical ablation study, confirming that the inclusion of the 3D structural priors provided by our dataset significantly enhances the model's ability to distinguish true signals from noise.

---

### Official Review · Reviewer_z91M · 2025-11-01

**Soundness:** 3
**Presentation:** 3
**Contribution:** 1
**Rating:** 2
**Confidence:** 4

**Summary:**

This paper presents CA-DEL, a new dataset for activity prediction using DNA-encoded libraries (DELs). The dataset includes results from screens against three homologous carbonic anhydrase targets sourced from CAS-DEL and DOS-DEL public libraries. Notably, the dataset contains 3D structures of all ligands in the form of binding poses predicted using the molecular docking tool SMINA. Several metrics are defined to evaluate machine learning models, mainly focusing on the accurate prediction of top compounds. An external testing set from ChEMBL is used to assess the models' ability to generalize to other molecular spaces. The benchmark includes a variety of models, from simple predictors based on molecular descriptors to recent 3D-based models such as DEL-Dock and DEL-Ranking.

**Strengths:**

- DEL data presents challenging problems for molecular modeling and provides numerous data points within a single experimental screen. The presented CA-DEL dataset combines public libraries, totaling over 350,000 training data points.
- The dataset includes three homologous targets, enabling users to evaluate how well the models can identify selective compounds.
- Binding poses of the ligands will be provided. These poses were generated using the docking protocol described in Section 3.2, which appears reasonable and utilizes more than one crystal structure per target.
- The choice of evaluation metrics is carefully considered and tailored to the hit identification task.
- Recent 3D models like DEL-Dock and DEL-Ranking are already part of the benchmark.
- The description of the dataset is clear, and data columns are explained in the appendix.

**Weaknesses:**

- The dataset is not available yet, which makes it impossible to review its quality. Since this is a paper in the "datasets and benchmarks" area, the dataset should be shared for review. I believe this is the most critical point because the dataset is the sole contribution of this paper (no new models or algorithms are introduced). We need to verify if the data is accessible, of high quality, and usable by other researchers.
- The Authors emphasize that this is the first DEL dataset containing 3D poses, but the KinDEL dataset also contains 3D poses predicted with molecular docking.
- It is unclear what the shaded background in Figure 2 means. It seems that these ranges should correspond to the Lipinski rules (MW<500), but why is the upper bound of QED not set at 1 (perfect QED score)? The figure caption mentions LogP, but this figure does not contain this information.
- The details on the source libraries are missing. There is no citation of CAS-DEL, DOS-DEL, and ChEMBL in the section on dataset construction. These sources should be cited, and their licenses should be discussed as CA-DEL is a combination of these datasets.
- In Section 3.3.2, the Authors say that they "decouple model performance from known unreliability of docking scoring functions" by evaluating their docking protocol by re-docking analysis. However, re-docking only one molecule per structure does not guarantee good docking poses and scores in general. Re-docking is rather a standard practice to see if the original ligand can be re-docked in the same place of the pocket of a protein in the exact same conformation that was observed for this ligand in the experimental structure.
- The text says that Sp and SubSp of the simple physicochemical baselines approach zero, demonstrating no meaningful predictive capacity. However, in Table 2 for 5doh and 3p3h, "Mol Weight" achieves Sp of -0.25 and SubSp of -0.125, which is not close to zero and very close to the best methods in this table (within the reported standard deviation). In other cases, the Benzene baseline also produces competitive correlation, for example, for 2hkf and 5fl4.
- The caption of Figure 4 should be more detailed to allow readers to understand the plot without reading the whole paper.
- I do not understand the motivation behind the zero-shot experiment. These are different proteins, so predictions of a model trained on one protein's data and applied to the other will not work very well. A detailed analysis should be performed to compare binding pockets and estimate the extent to which the same ligands can bind to both pockets. I suspect that only non-selective ligands can be discovered in this manner.
- A few typos should be corrected, e.g. "Tabel 2 and Tabel 3"
- In Figure A5, I do not see green sticks that are mentioned in the caption, only the cyan ligand.

**Questions:**

1. Where do you plan to publish the dataset and under what license? Will it follow FAIR (Findable, Accessible, Interoperable, Reusable) principles?
2. The Authors say that they deliberately chose testing molecules with a significant distributional shift to simulate the progression to optimized lead compounds. Later, the Authors say that their benchmark "directly evaluates a model's capacity to guide lead optimization." However, I am unsure if such a model trained on noisy DEL data can be useful for lead optimization, where structural changes may be very small and the activity is rather nuanced. Furthermore, the evaluation metrics are focused on top molecules, which simulates hit identification more than lead optimization. Do you expect the models that solve your benchmark to be good at both hit identification and lead optimization?
3. Section 4.3 concludes with "The experimental results on the CA-DEL dataset show consistent performance improvements [...], qualitatively validating the CA-DEL dataset’s effectiveness from an information-theoretic perspective." Could you clarify what information-theoretic perspective you had in mind?

---

> ### Author Response · Authors · 2025-11-27
> **Rebuttal by Authors**
>
> Thank you very much for your reply! Regarding your concerns, our responses are below:
> ### Weakness 1
>
> **Reviewer Comment:**
> > The dataset is not available yet, which makes it impossible to review its quality. Since this is a paper in the "datasets and benchmarks" area, the dataset should be shared for review. I believe this is the most critical point because the dataset is the sole contribution of this paper (no new models or algorithms are introduced). We need to verify if the data is accessible, of high quality, and usable by other researchers.
>
> **Response:**
> We fully agree that accessibility is paramount for a benchmark paper. We have uploaded the complete processed dataset, along with the raw docking files (to ensure reproducibility), to Zenodo (https://zenodo.org/records/17656024). The dataset is released under the CC BY 4.0 license, complying with FAIR principles to ensure it is Findable, Accessible, Interoperable, and Reusable for the research community.
>
> ---
>
> ### Weakness 2
>
> **Reviewer Comment:**
> > The Authors emphasize that this is the first DEL dataset containing 3D poses, but the KinDEL dataset also contains 3D poses predicted with molecular docking.
>
> **Response:**
> Thank you for the correction. We acknowledge the contribution of KinDEL. Our intended claim was that CA-DEL is the first 3D-structural dataset specifically for the Carbonic Anhydrase (CA) family, which presents different structural challenges compared to kinases. We have revised the manuscript to clarify that CA-DEL complements existing works like KinDEL by expanding the structural coverage to this new protein family, rather than claiming to be the first 3D DEL dataset globally.
>
> ---
>
> ### Weakness 3
>
> **Reviewer Comment:**
> > It is unclear what the shaded background in Figure 2 means. It seems that these ranges should correspond to the Lipinski rules (MW<500), but why is the upper bound of QED not set at 1 (perfect QED score)? The figure caption mentions LogP, but this figure does not contain this information.
>
> **Response:**
> We appreciate the detailed observation.
>
> * **Shaded Areas:** The light blue areas represent the 10th and 90th percentiles of FDA-approved oral new chemical entities, as reported by Shultz et al. (2018), representing "drug-like" property space.
> * **QED:** We have removed the shading for QED. Unlike the physicochemical properties included in Lipinski's rules (e.g., LogP, MW), QED is a composite score ranging from 0 to 1. Therefore, applying the percentile distribution shading is not appropriate in this context.
> * **LogP:** We have updated the figure and caption to explicitly include and label the LogP distribution. We have updated the caption to include the citation: Shultz, M. D. (2019). Setting expectations in lead optimization. Bioorganic & Medicinal Chemistry Letters.
>
> ---
>
> ### Weakness 4
>
> **Reviewer Comment:**
> > The details on the source libraries are missing. There is no citation of CAS-DEL, DOS-DEL, and ChEMBL in the section on dataset construction. These sources should be cited, and their licenses should be discussed as CA-DEL is a combination of these datasets.
>
> **Response:**
> We have added the necessary citations and license descriptions for all source libraries (CAS-DEL, DOS-DEL, and ChEMBL) in the "Dataset Construction" section to ensure full attribution and transparency.
>
> * **CAS-DEL:** Rui Hou, Chao Xie, Yuhan Gui, Gang Li, and Xiaoyu Li. Machine-learning-based data analysis method for cell-based selection of dna-encoded libraries. ACS omega, 2023.
> * **DOS-DEL:** Christopher J Gerry, Mathias J Wawer, Paul A Clemons, and Stuart L Schreiber. Dna barcoding a complete matrix of stereoisomeric small molecules. Journal of the American Chemical Society, 141(26):10225–10235, 2019.
> * **ChEMBL:** Anna Gaulton, Louisa J Bellis, A Patricia Bento, Jon Chambers, Mark Davies, Anne Hersey, Yvonne Light, Shaun McGlinchey, David Michalovich, Bissan Al-Lazikani, et al. Chembl: a large-scale bioactivity database for drug discovery. Nucleic acids research, 40(D1):D1100–D1107, 2012.
>
> ---

---

> > ### Author Response · Authors · 2025-11-27
> > **Rebuttal by Authors**
> >
> > ### Weakness 5
> >
> > **Reviewer Comment:**
> > > In Section 3.3.2, the Authors say that they "decouple model performance from known unreliability of docking scoring functions" by evaluating their docking protocol by re-docking analysis. However, re-docking only one molecule per structure does not guarantee good docking poses and scores in general. Re-docking is rather a standard practice to see if the original ligand can be re-docked in the same place of the pocket of a protein in the exact same conformation that was observed for this ligand in the experimental structure.
> >
> > **Response:**
> > We acknowledge that re-docking a single molecule does not guarantee global accuracy. However, given that the Carbonic Anhydrase binding pocket is structurally conserved and well-studied, we utilized re-docking as a validation of our protocol's feasibility rather than a claim of perfect accuracy for every ligand. To mitigate sampling uncertainty, our dataset provides the top-9 docking poses for each molecule, allowing users to perform ensemble analysis or filter based on interaction patterns.
> >
> > ---
> >
> > ### Weakness 6
> >
> > **Reviewer Comment:**
> > > The text says that Sp and SubSp of the simple physicochemical baselines approach zero, demonstrating no meaningful predictive capacity. However, in Table 2 for 5doh and 3p3h, "Mol Weight" achieves Sp of -0.25 and SubSp of -0.125, which is not close to zero and very close to the best methods in this table (within the reported standard deviation). In other cases, the Benzene baseline also produces competitive correlation, for example, for 2hkf and 5fl4.
> >
> > **Response:**
> > Thank you for this keen observation. We agree that the baselines are not "near zero" in all cases.
> >
> > * **Phenyl Baseline:** The competitive performance here is expected, as the sulfonamide group (often attached to phenyl rings) is the key pharmacophore for CA inhibition.
> > * **MW Baseline:** The performance of Molecular Weight (-0.25) and (-0.125) suggests that non-specific binding driven by size or hydrophobicity introduces a systematic bias in the experimental selection conditions. We have revised the text to accurately reflect that these baselines are competitive in specific subsets, which highlights the difficulty of the benchmark.
> >
> > ---
> >
> > ### Weaknesses 7, 9 & 10
> >
> > **Reviewer Comment:**
> > > The caption of Figure 4 should be more detailed to allow readers to understand the plot without reading the whole paper.
> > > A few typos should be corrected, e.g. "Tabel 2 and Tabel 3".
> > > In Figure A5, I do not see green sticks that are mentioned in the caption, only the cyan ligand.
> >
> > **Response:**
> > * **Figure 4:** We have expanded the caption to be self-explanatory.
> >     > Figure 4: Model performance evaluation using the Top-N hit rate on the CA-DEL dataset. This plot shows the percentage of high-affinity ”hits” (defined as the top 5% of binders) successfully identified within the top 200 predictions of each model’s ranked list. This metric quantifies the practical utility of a model in resource-constrained screening campaigns.
> > * **Typos:** We have corrected "Tables 2 and 3" and other minor errors.
> > * **Figure A5:** We clarified in the caption that both the cyan ligand and the "green" reference refer to the same ligand in different views (global vs. zoomed-in interaction view) to avoid confusion.
> >     > Figure A5: Validation of the docking protocol via re-docking. (Left) Overall view of the computationally generated binding pose (ligand shown in cyan sticks) within the protein’s active site. (Right) A zoomed-in view detailing the critical interactions. The computationally generated pose (cyan) accurately reproduces the experimental binding mode(, such as zinc chelation and hydrogen bonds), forming key interactions with the active site residues (pink sticks).
> >
> > ---
> >
> > ### Weakness 8
> >
> > **Reviewer Comment:**
> > > I do not understand the motivation behind the zero-shot experiment. These are different proteins, so predictions of a model trained on one protein's data and applied to the other will not work very well. A detailed analysis should be performed to compare binding pockets and estimate the extent to which the same ligands can bind to both pockets. I suspect that only non-selective ligands can be discovered in this manner.
> >
> > **Response:**
> > The motivation for the zero-shot experiment lies in the high structural conservation of the Carbonic Anhydrase active site (specifically the Zinc-binding histidines) across isoforms (CA2, CA9, CA12). This setup tests whether the model learns the shared physical interaction physics (e.g., metal coordination, hydrogen bonding) rather than simply memorizing ligand identity.
> >
> > ---

---

> > > ### Author Response · Authors · 2025-11-27
> > > **Rebuttal by Authors**
> > >
> > > ### Question 1
> > >
> > > **Reviewer Comment:**
> > > > Where do you plan to publish the dataset and under what license? Will it follow FAIR (Findable, Accessible, Interoperable, Reusable) principles?
> > >
> > > **Response:**
> > > The dataset has been uploaded to Zenodo under the CC BY 4.0 license (https://zenodo.org/records/17656024). This repository ensures the data is Findable (via DOI), Accessible (open download), Interoperable (standard .sdf and .pdb formats), and Reusable (clear licensing), strictly adhering to FAIR principles.
> > >
> > > ---
> > >
> > > ### Question 2
> > >
> > > **Reviewer Comment:**
> > > > The Authors say that they deliberately chose testing molecules with a significant distributional shift to simulate the progression to optimized lead compounds. Later, the Authors say that their benchmark "directly evaluates a model's capacity to guide lead optimization." However, I am unsure if such a model trained on noisy DEL data can be useful for lead optimization, where structural changes may be very small and the activity is rather nuanced. Furthermore, the evaluation metrics are focused on top molecules, which simulates hit identification more than lead optimization. Do you expect the models that solve your benchmark to be good at both hit identification and lead optimization?
> > >
> > > **Response:**
> > > This is an insightful question. We acknowledge that our current evaluation (Top-K retrieval) aligns more closely with Lead Discovery (LD). However, we argue that effective denoising is the prerequisite for Lead Optimization. DEL data is inherently noisy; by training models to distinguish true binders from noise (even if the signal is subtle), we lay the foundation for fine-grained SAR (Structure-Activity Relationship) analysis. While CA-DEL primarily benchmarks discovery capabilities, it provides the necessary structural "ground truth" (via docking) to begin developing models capable of the subtle distinctions required for optimization.DEL data is inherently noisy. A model that can successfully retrieve "shifted" test molecules (which represent optimized leads) from this noise has demonstrated that it is not merely memorizing training hits, but has learned generalizable Structure-Activity Relationships (SAR). Therefore, we do expect models that solve this benchmark to excel in both tasks: they perform Hit Identification by filtering noise, and they support Lead Optimization by correctly generalizing activity predictions to novel, optimized chemical spaces—a capability that simple memorization cannot achieve.
> > >
> > > ---
> > >
> > > ### Question 3
> > >
> > > **Reviewer Comment:**
> > > > Section 4.3 concludes with "The experimental results on the CA-DEL dataset show consistent performance improvements [...], qualitatively validating the CA-DEL dataset’s effectiveness from an information-theoretic perspective." Could you clarify what information-theoretic perspective you had in mind?
> > >
> > > **Response:**
> > > By "information-theoretic perspective," we refer to the concept of Information Gain. Qualitatively, a dataset containing 3D structural priors provides more relevant information (reduces entropy) regarding the binding mechanism than 1D/2D representations alone. The fact that our models achieve better performance when utilizing these 3D features indicates that the CA-DEL dataset successfully encodes this additional, high-value information, thereby validating its effectiveness.

---

### Official Review · Reviewer_tF48 · 2025-11-02

**Soundness:** 3
**Presentation:** 3
**Contribution:** 3
**Rating:** 8
**Confidence:** 2

**Summary:**

This work introduces CA-DEL, a comprehensive, open benchmark for learning from DNA-Encoded Library (DEL) screening data. The benchmark covers 3 homologous carbonic anhydrase isoforms (CAII, CAIX, and CAXII) and integrates both 2D molecular structures and 3D protein–ligand conformations. It also provides experimentally measured binding affinities for validation, enabling direct evaluation of model generalization from noisy DEL read counts to true biological activity. The dataset includes realistic out-of-distribution splits and validated 3D docking ensembles, addressing key gaps in existing DEL benchmarks that lack multi-target or structure-aware components.

**Strengths:**

CA-DEL is a well-designed and rigorously curated benchmark that directly responds to the community’s need for public DEL datasets with ground-truth validation. The authors demonstrate the dataset’s utility through extensive baseline experiments, showing that 3D geometric learning methods outperform 2D and classical baselines in both correlation and Top-N hit rate analyses. The focus on isoform selectivity and out-of-distribution generalization is scientifically meaningful and reflects real-world challenges in drug discovery.

**Weaknesses:**

The analysis step shown in Figure 1 to get 3D structure information from 3D complex generations is not clearly explained in the manuscript.

**Questions:**

Are there plans to expand CA-DEL to additional protein families?

---

> ### Author Response · Authors · 2025-11-27
> **Thank you very much for your reply! Regarding your concerns, our responses are below:**
>
> ### Weakness 1
>
> **Reviewer Comment:**
> > The analysis step shown in Figure 1 to get 3D structure information from 3D complex generations is not clearly explained in the manuscript.
>
> **Response:**
> Thank you for pointing out the ambiguity regarding the "analysis" step in Figure 1. We clarify that this step specifically refers to the reliability verification of our docking process, where we conduct detailed case studies of the binding modes to ensure the generated complexes are biologically plausible.
>
> A concrete example of this analysis is presented in **Figure A5**, which illustrates the specific interactions and pose reliability of the generated complexes. We apologize if this connection was not immediately clear.
>
> In the revised manuscript, we will update the caption of Figure 1 as follows:
>
> ```latex
> \caption{\textbf{Schematic overview of the proposed structure-based deep learning framework for DEL analysis.}
> The workflow is organized into three distinct phases:
> (1) \textbf{Data Screening}, where raw DEL screening data is processed to align target Uniprot IDs with ligand SMILES and affinity metrics;
> (2) \textbf{Conformation Generation}, which utilizes molecular docking to generate 3D protein-ligand complexes. An intermediate \textit{analysis} step is employed to evaluate the docking results and verify the correctness of the binding conformations before finalizing the 3D structure information;
> and (3) \textbf{Information Extraction}, where the validated structural features are fed into a neural network model to predict bioactivity scores and evaluate compound labels.}
> ```
>
> ---
>
> ### Question 1
>
> **Reviewer Comment:**
> > Are there plans to expand CA-DEL to additional protein families?
>
> **Response:**
> Thank you for this valuable question. Expanding CA-DEL to additional protein families is highly feasible and is a key part of our future roadmap.
>
> The CA-DEL framework is designed as a standardized workflow that is agnostic to the specific protein target. As long as enrichment values and defined protein-ligand pairs are available, we can utilize molecular docking or advanced structure prediction tools (such as AlphaFold 3) to generate the necessary structural inputs.

---

### Note · Program_Chairs · 2026-01-17
**Submission Desk Rejected by Program Chairs**

The following references in this submission do not refer to real documents and/or have major errors in bibliographic information:

 Hao Jiang, Connor W Coley, and William H Green. Advances in the application of molecular docking for the design of dna-encoded libraries. Journal of chemical information and modeling, 55(7):1297-1308, 2015.
Jonathan M Stokes, Kevin Yang, Kyle Swanson, Wengong Jin, Andres Cubillos-Ruiz, Nina M Donghia, Craig R MacNair, Shawn French, Lindsey A Carfrae, Zohar Bloom-Ackermann, et al. Deep learning-based prediction of protein-ligand interactions. Proceedings of the National Academy of Sciences, 117(32):19338-19348, 2020.
Eric J Martin, Valery R Polyakov, Lan Zhu, and Lu Zhao. Quantitative analysis of molecular recognition in dna-encoded libraries. Journal of chemical information and modeling, 57(9):2077-2088, 2017.